# Towards Practical Conservation Cloning: Understanding the Dichotomy Between the Histories of Commercial and Conservation Cloning

**DOI:** 10.3390/ani15070989

**Published:** 2025-03-29

**Authors:** Ben J. Novak, Stewart Brand, Ryan Phelan, Sasha Plichta, Oliver A. Ryder, Robert J. Wiese

**Affiliations:** 1Revive & Restore, 1505 Bridgeway #203, Sausalito, CA 94965, USA; 2Department of Chemical and Biological Engineering, University of British Columbia, Vancouver, BC V6T 1Z4, Canada; 3Beckman Center for Conservation Research, San Diego Zoo Wildlife Alliance, Escondido, CA 92027, USA; 4North Carolina Museum of Natural Sciences, Raleigh, NC 27601, USA; bob.wiese@naturalsciences.org

**Keywords:** genetic rescue, cloning, somatic cell nuclear transfer, black-footed ferret, Przewalski’s horse

## Abstract

While cloning for laboratory and commercial applications has become routine, research relevant to wildlife conservation predominantly comprises “one-off” studies. Reviews of conservation cloning have led many authors to conclude that cloning is not yet a viable tool to aid the management and recovery of endangered species. Major milestones achieved recently for Przewalski’s horse and the black-footed ferret challenge this perception. To understand why the perception that cloning is not a viable conservation tool has become so pervasive, we evaluated previous reviews and attempted to fill major knowledge gaps. We provide for the first time a complete history of species successfully cloned via nuclear transfer methods with attention to fertility, reproduction, and longevity of clones. For conservation-related studies, we sent inquiries to scientists to obtain unreported information, including the reasons their studies were initiated and terminated. The major reasons cited for the short duration and small scales of conservation cloning research were not biological, but resource dependent, particularly funding limitations. As demonstrated for Przewalski’s horse and the black-footed ferret, conservation cloning is worth investment in the present and can be used to achieve genetic diversity management goals that no other method can currently provide.

## 1. Introduction

The notion of using cloning, via somatic cell nuclear transfer, to aid the conservation of wild taxa was first considered in the 1980s among conservationists preserving viable cell cultures of wild taxa in liquid nitrogen, sometimes called “frozen zoos”. Kurt Benirschke, the founder of the San Diego Zoo Wildlife Alliance’s Frozen Zoo^®^, alluded to the use of cryopreserved cells and advanced reproductive technologies for conservation in 1984: “They [cell lines] are preserved for the future when it will be useful to reconstruct pedigrees or when animals of this nature will no longer be available” [1]. These are prescient words, given that at the time, advanced reproductive technologies could only reproduce individuals from germ cells, pluripotent embryonic cells, and fetal cells for most species, while the vast majority of cells and tissues collected and cryopreserved for wild taxa were adult somatic cells. To this day, the majority of biosamples collected from wild species are adult, often post-mortem somatic tissues. However in 1996, 22 years after Dr. Benirschke’s publication, the cloning of Dolly the sheep from adult somatic cells enabled a major expansion of cloning capabilities in both research and practical applications, opening the door for the use of cloning in conservation [2]. Since that time, a number of conservation-related cloning research efforts ensued; however, it has taken nearly 30 years for cloning to achieve an actual conservation application to aid the recovery of a species. The first successful applied conservation outcome from cloning was announced in 2024 with the birth of two kits from the clone of an endangered species, the black-footed ferret (*Mustela nigripes*) [3]. The birth of the kits marks the first time that a clone of an endangered species has reproduced as part of a conservation breeding program. However, more significantly, the mother of the kits is a clone of an individual that died in 1988 with no descendants [4]. Until the birth of the clones and their subsequent offspring, all living black-footed ferrets descended from just seven founders; the successful breeding of the clone therefore introduces an eighth founder to the population and restores lost allelic diversity [5]. While artificial insemination has restored lost alleles from previous generations using cryopreserved semen within a breeding program [6], to our knowledge, this is the first time the alleles of an entirely unrepresented individual have been revived and integrated in any conservation breeding population. It demonstrates a clear precedent for the use of cloning in the genetic management of populations, especially for species with limited genetic diversity that rely on the maintenance of small populations for many generations while conservationists work to restore habitat and mitigate threats in the wild.

Since the cloning of Dolly the sheep, research into cloning domestic and model research organisms has also diversified, but, in sharp contrast to conservation, has produced a series of notable applied contributions to multiple industries across the globe. Cloning technology has become commercialized by several companies in multiple nations (including US, India, Brazil, Argentina, UAE, Korea, and China), which today perform cloning services for improved selective breeding and the reproduction of valuable domestic livestock (e.g., cattle for meat, water buffalo for milk), animal competition (e.g., rodeo bulls, polo horses, beauty camels), service animals (e.g., drug detecting, rescue, and guide dogs) and companion animals (e.g., dogs and cats) [7,8,9,10,11,12]. Cloning has even been used to help save rare cattle breeds from extinction, such as Enderby, Zhangmu, and Apeijiaza breeds [13,14,15].

Cloning advances and applications have been continually and extensively reviewed for both domestic and wild taxa, including an entire Special Issue published recently to commemorate advancements in the field in the 25 years after the birth of Dolly [16,17,18,19,20,21,22]. Reviews of the history of cloning domestic taxa reveal continual diversification of scientific and commercial applications [19,23,24,25,26,27,28]. On the other hand, reviews concerning conservation-relevant cloning efforts have consistently characterized cloning as a technology that is not yet an effective or justified tool for applied conservation. These conclusions are primarily based on the low efficiency rates of cloning and animal welfare concerns [29,30,31,32,33,34].

In this review, we analyze the history of cloning in an attempt to understand how the discordance between views of cloning for conservation and cloning for other industries have arisen and revisit the barriers that have so far prevented cloning from being a more widely applied conservation tool. From this perspective, we offer suggestions of how conservation cloning efforts may emulate the cloning research and applications of domestic taxa. Firstly, we found that all previous reviews of the history of cloning presented mismatched datasets and histories pertaining to which species have been cloned. Therefore, we first scoured peer-reviewed and gray literature to exhaustively identify the most comprehensive list of species and subspecies (hereon referred to collectively as taxa) that have been successfully cloned (defining success as achieving a live birth or hatching of a cloned individual). Understanding the history and perceptions of cloning also required as much information as could be gained on (1) the life-history milestones of clones (i.e., reaching adulthood, reproduction, lifespans) and (2) assessing the motivations for initiating and terminating cloning research. We gleaned this information from peer-reviewed and gray literature as much as possible, and where this information was absent, we sent inquiries directly to scientists that were involved in conservation-relevant cloning efforts.

## 2. The Complete History of Successfully Cloned Animals

In this paper, we use the term cloning to refer specifically to nuclear transfer, a process by which an organism is cloned by inserting the nucleus of a donor diploid cell (pluripotent or somatic) cell into an enucleated oocyte (egg cell, aka cytoplast) to create and stimulate the development of an embryo. We do not review alternate methods of cloning, such as embryo splitting [34,35,36].

While cloning became a well-known advanced reproductive technology in 1996 with the famous cloning breakthrough of Dolly the sheep, the technology has a much deeper and more complex history, with supporting research stemming back to the early 1900s [24]. The first successful cloning milestone was achieved in 1952, which produced embryos of the northern leopard frog (*Rana pipiens*) by transferring donor nuclei into recipient oocytes [37]. This established the protocols that would yield the first cloned organisms, two taxa of frog, in 1957 [38,39] (Table 1). In the nearly 70 years that have followed, cloning research has been performed using donor cells and oocytes of dozens of taxa, ranging from fruit flies to whales [24,33,40]. The vast majority was conducted for cellular and developmental research with no attempts to produce live offspring [41]. While this research provides many insights about mito-nuclear and nuclear-epigenetic interactions between divergent species, cloning research needs to produce at least 1 live birth or hatching to be considered a potential reproductive technology for a given taxon. To our knowledge, the 56 species and subspecies listed in Table 1 represent the first exhaustively comprehensive list of taxa in which cloning research yielded live births to date, incorporating data from scientific and gray literature.

For some research purposes, producing a live birth is a sufficient milestone to validate the potential use of cloning for a given taxon, but for most commercial and all conservation applications clones must reach sexual maturity and reproduce. Clone longevity, fertility, and reproductive potential has not been widely reported. Still, we found reports or received updates through personal communication regarding reproductive capability for 42 taxa (Table 1). Of these 42 taxa, failure to produce fertile clones was reported for only 2, the common frog (*Rana temporaria*) and banteng (*Bos javanicus*) [48,77]. Adult clones of an additional two taxa have been produced too recently to yet reach sexual maturity (Rhesus macaque, *Macaca mulatta*, [63] and Przewalski’s horse, *Equus przewalskii* [102]), and therefore their fertility cannot yet be assessed. Of the remaining 12 taxa, one effort produced a single clone that reached weaning age but died before sexual maturity (European mouflon, *Ovis aries musimon*). Clones for five taxa died before weaning [47,69,70,76,84,97]. Healthy adult clones were reported for six taxa, but no reports of fertility could be found or obtained [46,48,51,71,86,87,103]. It is possible that clones of some or all of those taxa were fertile and possibly reproduced. When assessing success rates of fertility in clones, it is not appropriate to include taxa for which fertility assessments could not be obtained or could not be conducted due to premature deaths. At face value, the success rate of producing fertile clones is therefore 40 of 42 taxa to date (95%), rather than 40 of 56 (71%) (Table 1). A caveat to this assessment is that the two cases in which clones were infertile do not establish that fertile clones cannot be produced for those taxa. Only a single viable clone was produced for the Banteng and no further efforts were pursued. Almost all cloning efforts that reported fertile clones produced more than one live birth, and most produced multiple sexually mature adults. It is likely that further cloning efforts would produce fertile Banteng clones. More intriguing is the case of common frog (*R. temporaria*) clones, which were produced by cross-species cloning using the oocytes of *R. japonica*. Interestingly, the inverse of *R. japonica* clones produced using *R. temporaria* oocytes produced fertile adults [48]. Same-species cloning was never reported for *R. temporaria*. It is possible that fertile *R. temporaria* clones could be obtained using same-species oocytes or oocytes from a mito-nuclear or epigenetically compatible taxon.

Interestingly, when comparing outcomes for cloning efforts for wild versus domestic taxa, the majority of successfully cloned taxa have been wild—33 of 56 (58%) (see Table 1). Many of those taxa were cloned to study embryology, specifically the dynamics of cellular reprogramming. Only 14 wild and 1 domestic taxa have been successfully cloned for the purpose of conservation research or applications, all of which have been mammals (Table 2). To understand the history of conservation-motivated cloning efforts, we expanded our search for publications in which the intent to produce live births was clear independent of the outcome. We found an additional 14 taxa for which producing live offspring was attempted and failed, including 3 fish taxa and 1 amphibian taxon (Table 2).

### 2.1. Measuring Success, and Optimizing Efficiency of Cloning Efforts

The relative success or failure of the cloning process is measured over incremental milestones: (1) the production of viable blastocysts, (2) successful incubation or post-implantation pregnancy, (3) full term development and birth or hatching of cloned offspring, (4) survival of clones past weaning, (5) reaching sexual maturity (for which success is measured by fertility), and (6) longevity. The majority of publications report results only through the perinatal stages, which does not necessarily encompass weaning. Each successive stage is reported with decreasing frequency, total longevity being the least reported aspect of clone life histories in peer-reviewed literature. Cloning exhibits the lowest live birth efficiency rates for any advanced reproductive technology. However, published efficiencies should be scrutinized when comparing different cloning efforts. There is no standard method for calculating efficiency results in cloning publications. For example, the efficiency of cloning Dolly can be calculated at 0.36% or 3.4% depending on whether unused viable embryos are included in the calculation [122]. It is most common to report efficiency as the number of live births produced from the number of embryos transferred or incubated for eventual birth/hatching, not the total number of embryos created *in vitro*. In this review we have calculated efficiencies for conservation related efforts by this metric. Published efficiencies across taxa average <1–3%.

Several factors that impact efficiency have been identified [27,123,124]: (1) incomplete or corrupted reprogramming of the donor genome due to cytoplasmic and nuclear epigenetic factors, (2) compatibility between the donor nuclear genome and oocyte mitochondrial genome (mito-nuclear compatibility), (3) donor cell type (e.g., fibroblasts, cumulus cells, stem cells, etc.), and (4) donor cell quality (affected by organismal age, cell culture conditions, etc.). Continued research in domestic taxa has shown that cloning efficiency can be significantly elevated by improvements to these parameters. Recent experiments to upregulate or downregulate certain epigenetic factors of donor somatic cells and/or oocytes have greatly increased efficiency rates in mice, cattle, and pigs [27]. Cloning efficiency was increased from 1–3% to 23.5% for mice, 7% to 23.6% for cattle, and 0.3% to 2.07% for pig [125,126,127]. Such optimization and experimentation have been rare in wild taxa. In conservation related efforts efficiencies have been reported at <1–18% (Table 2), but the low numbers of embryos and live births in these programs diminish statistical confidence when compared to the much larger sample sizes for related domestic taxa.

While potential problems with mito-nuclear compatibility and epigenetic factors have received a great deal of attention in published reviews, individual donor cell line quality and genotype has been reported as a contributor to results [128]. For all taxa for which data was reported from different donor cell lines, oocyte donors, or experimental treatments, variation in cloning outcome is evident. For example, when cloning gaur (*Bos gaurus*) Srirattana et al. [70] reported an overall cloning efficiency of 2.4% from two cell lines, each of which received two different treatments. However, the efficiency varied from 0–11% between the two cell cultures and treatments. The reasons why particular donor cell lines perform better than others have not been examined. Low sample size in general is an issue for conservation-related cloning, which compromises the ability to draw statistically significant conclusions for any aspect of the cloning process.

Cell lines that yield viable, healthy clones have been reported to produce consistent results repeatedly. The same donor cells used to clone Dolly were used to produce additional clones in later studies [129]. Sinclair et al. [129] reported higher efficiency from the same technique used to clone Dolly, as well as from newer experimental treatments (13% and 20% live births, respectively, vs. the 3.4% success rate of Dolly).

Viable donor cell lines can also be recloned, meaning that donor cells can be obtained from adult clones and used to clone the original donor line again [130,131]. Most notably, a single mouse donor cell line was recloned for 25 generations; longevity and fertility did not vary significantly after each subsequent cloning [130].

Donor cell lines that do not support normal development of clones are not necessarily without useful application. Serial nuclear transfer, also known as donor cell “rejuvenation,” is a process by which adult donor cells are used to produce cloned embryos, which are terminated before reaching full term. Terminated cloned embryos or fetuses are then used as somatic cell donors for a second round of cloning, taking advantage of the higher rates of efficiency achievable with embryonic or fetal cells in order to replicate the original adult cell donor. At least, that is the hypothesis behind the process; mixed results have been obtained, with increased efficiency achieved in some experiments [132,133] but not others [66].

Serial nuclear transfer has been applied to amphibians in attempts to overcome developmental arrests of clones from adult donor cells [134]. To date, healthy adult cloned amphibians have not been produced from adult somatic donor cells. Amphibian embryos cloned from adult donor cells typically arrest at the gastrula stage or earlier. However, multiple studies, as reviewed by Gurdon [134], produced larvae that reached the “heart-beat” stage, some of which obtained these further developmental stages by applying serial nuclear transfer. Gurdon reported obtaining swimming-stage tadpoles from adult donor cells, with a single clone surviving just past metamorphosis [135]. Therefore, it should be possible to obtain a healthy adult amphibian clone from adult donor cells. However, since the 1980s, when mammal cloning was achieved, amphibian cloning research has declined dramatically, with only one current effort to advance cloning for a conservation purpose: to recover the Gastric Brooding frog (*Rheobatrachus silus*) (personal communication, Dr. Michael Archer).

### 2.2. Health, Fertility, and Longevity of Clones

Of the 56 cloned taxa listed in Table 1, healthy adult clones were reported for 50 taxa (89%), while failure to produce healthy adult clones was reported for only five taxa (9%) [47,69,70,76,84,97]. The causes of death in these cases include developmental abnormalities, infection, and complications during care. In each of these five cases, researchers used a method known as interspecies, or cross-species cloning, a method which uses oocytes from different taxa from the somatic donor cell.

Developmental abnormalities resulting in perinatal deaths or poor health are reported at higher frequencies in cloned organisms than individuals born through other advanced reproductive technologies [136]. However, to date, there have been no developmental abnormalities uniquely observed in clones; all abnormalities observed in clones have been observed in natural-born animals of the same or related taxa. Further, it remains unknown what proportion of perinatal deaths stem from problems in the cloning process or non-cloning related factors, even among cloning studies of domestic taxa for which large numbers of clones have been generated.

Causes of early death after the perinatal period have yet to be linked to abnormalities resulting from cloning issues. Research in domestic taxa shows that clones that survive the perinatal period can be expected to attain normal life expectancies, have the same resilience/susceptibility to disease as natural-born individuals, and produce healthy offspring even if the cloned parent was born with abnormalities [10,129,136,137,138]. Long-term reporting on clone longevity in peer-reviewed literature is scarce, but reports of cloned cattle, cats, and mice exceeding normal life expectancy can be found in news articles [139,140,141]. As reported in the peer-reviewed literature, the four identical cloned sisters of Dolly the sheep all reached old age, exceeding Dolly’s lifespan. Only one of those clones exhibited the same osteoarthritis developed by Dolly [129], and it was reported to be less severe. These four clones were genetically identical to Dolly, but were raised in an outdoor pasture while Dolly lived indoors, indicating that the disease was most likely influenced by environmental factors (e.g., diet, air quality, or the ability to walk longer distances while grazing naturally). Reports of longevity, health, and causes of adult death for wild cloned taxa are virtually absent in peer-reviewed literature, but personal communication responses to our inquiries to cloning scientists and practitioners appear superficially to parallel trends for domestic taxa, with several clones of wild taxa reportedly reaching extreme old age (Table 3).

Of the 15 taxa successfully cloned for conservation research or application, 10 reportedly produced healthy adults. In total, evaluations of fertility and reports of reproduction were obtained for seven taxa, of which only two were published in peer-reviewed publications. Only one taxon, the banteng, was reported as infertile, while viable reproduction has been reported for five taxa. The two cloned Przewalksi’s horses born to date are not yet sexually mature to evaluate fertility. However, it should be noted that the effort that finally succeeded in producing a viable adult clone of an Argali (*Ovis ammon*) was not for conservation research nor application, but was for illegal trophy hunting, and has therefore designated in Figure 1 as a “commercial” cloning endeavor given the economic motivations of the illicit work [98,99]. Nonetheless, the work proves that Argali can be viably cloned should conservationists choose to continue cloning efforts. Given that the Argali was one of the first endangered taxa for which conservation-motivated research began, the life-history data for the illegally cloned Argali male is considered in light of its conservation relevance. Of the five taxa for which reproductive clones were produced, only one was cloned for a direct conservation application; the rest were for research purposes. Few efforts have ever been designed for conservation applications (Table 3). Only one other program, notably excluded from Table 2, was designed in a manner to advance research and simultaneously produce clones that would benefit the recovery of their wild populations: cloning efforts for the European mouflon [71,143] Table 3; however, the international Union for the Conservation of Nature (IUCN) no longer considers European mouflon a distinct subspecies. It has been regarded as a descendant of historically feral domestic sheep, and thus was reclassified from *Ovis orientalis (gmelini) musimon* to *Ovis aries musimon* [144,145]. European mouflon are no longer considered a distinct nor endangered taxon, nullifying the biodiversity conservation relevance of former genetic rescue driven research efforts. Although, as with many feral populations, conserving the ferally originating populations of European mouflon does have cultural and potentially ecological value to Mediterranean island habitats.

## 3. Discussion

### 3.1. Trends in Conservation-Related Cloning Efforts

While the history of conservation-related cloning efforts seems to support the conclusion that cloning is not ready yet for conservation applications, a closer examination of trends in conservation-relevant cloning efforts, (particularly in comparison with domestic taxa), suggests that further consideration is warranted. All cloning efforts that failed to produce live births (Table 2) or healthy adult clones (Table 1) share several common variables that may have impacted success. Each effort employed cross-species cloning and all but one effort performed far fewer experimental replicates than typical efforts that yielded healthy adults. One potential reason that cross-species cloning may be responsible for these relative failures is that cross-species cloning has exhibited lower efficiencies than same-species cloning. However, this may not be a strong pattern: (1) most efforts using cross-species cloning produced too few offspring to draw meaningful statistics and (2) there are few studies directly comparing same-species and cross-species cloning. It has been hypothesized that the evolutionary distance between the donor taxa and the oocyte taxa is a barrier to cross-species success, as successful mammalian cross-species cloning has only been achieved for taxa diverged <6.5 million years [41,123,124] Table 2. However, many published reviews overlooked the successes of cross-species fish and amphibian clones [39,43,46,47,48,49,51,54], for which multiple taxa have produced fertile clones using donor cell and oocyte taxa that diverged 9–106 million years ago (Table 4), suggesting that fish and amphibians may be more amenable to successful cross-species reproduction via cloning. Notably, the need to use surrogate mothers as ovum donors and/or gestational surrogates to clone mammals presents additional variables: even closely related mammalian taxa may have endocrine regulation, gestational lengths, and live birth weights/sizes that differ enough to inhibit the development of cross-species clones. A diversity of mammalian somatic/oocyte combinations have proven functionally compatible *in vitro;* rabbit oocytes in particular have shown to activate a wide range of genomes, supporting embryonic development to the blastocyst stage [41], and in some instances that pursued further development *in vitro* or *in vivo* later stages were achieved. Perhaps the most intriguing case was when Chen et al. [146] attempted to clone giant pandas (*Ailuropoda melanoleuca*) using domestic rabbit (*Oryctolagus cuniculus domestica*) oocytes and a domestic cat (*Felis catus*) surrogate mother. This combination of taxa was ~94 million years (panda/rabbit) and 55 million years (panda/cat) diverged [119]. While a live birth was not produced, the cloned embryo developed for approximately ten days in utero. It is not possible to determine how the variables of the donor cytoplast, uterine environment, or diet of the surrogate mother influenced the result. However, we believe the results support further study of cross-species cloning variables.

One variable that did not appear to be a barrier for the panda experiment was mito-nuclear incompatibility, which may not be as significant a barrier as many have hypothesized. The rabbit mitochondrial DNA was incompatible with the panda nuclear genome, but this may not have inhibited development, as it was observed that the panda donor mitochondrial DNA proliferated and the rabbit mtDNA disappeared, resulting in an embryo that possessed only panda mtDNA [146]. This same phenomenon was observed in viable cross-species clones of domestic carp, *Cyprinus carpio* var. *Wuyuanesis*, generated with oocytes of goldfish, *Carassius auratus*, diverged ~34 million years [119]. The clones possessed only domestic carp mtDNA [147].

Although cross-species cloning is the underlying method of all cloning efforts that failed to produce healthy adults, cross-species cloning has worked effectively for the majority of its applications to date. A total of 26 (81%) of the total 32 taxa for which cross-species cloning was used have reported healthy adult clones, of which 17 were confirmed fertile/reproductive (see Table 1). There are no strong trends when comparing and contrasting successes versus relative failures in cross-species cloning, indicating the factors influencing outcomes are complex, and as yet largely unknown. By our observation, we conclude the best predictor of cloning success to be scale (i.e., the number of attempts/treatments/replicates). Increasing scale is not simply a matter of increasing the number of replicates to compensate for low efficiency rates; it is multidimensional. Large-scale efforts allow researchers to compare multiple parameters, including cell culture conditions [111] or donor cell type [81]. They also afford the opportunity to test differing treatments to optimize efficiency [27]. In general, successful efforts performed more experimental replicates and tested more individual donor cell lines, subjected to multiple treatments, than failed efforts [110 for example]. Biological complications/barriers were not cited nor reported by personal communications as the primary reason that cloning efforts failed to produce healthy adult clones or were terminated for any taxon in recent decades, particularly for conservation-relevant efforts (Table 3).

What we found when examining the history of cloning is that the most significant differences between conservation-related efforts versus non-conservation cloning efforts were program persistence, refinement, and the continued use of cloning for a given taxon once success was achieved. We have illustrated this contrast in Figure 1. For conservation-related cloning efforts, proof-of-principle, i.e., achieving successful pregnancies or a single birth became the end-point for almost all programs, whereas for most domestic taxa, it was a launching point for independent laboratories and commercial entities to continue cloning hundreds and thousands of each respective taxon since (Figure 1). Conservation related cloning efforts collectively have produced only 38 adult clones from 10 taxa (Table 2), with clones of an eleventh taxon, the Przewalski’s horse clone, reportedly developing normally and nearing sexual maturity [102].

### 3.2. Factors Hindering Progress for Conservation Cloning

Several factors contribute significantly to the divergent trends in cloning for conservation versus cloning for research or commercial applications. The first and foremost is funding: cloning domestic taxa has an apparent return on investment, and funding support for model research overwhelmingly overshadows funds available for conservation [148]. The second is that reproductive biology, husbandry, and welfare knowledge of domestic taxa vastly exceeds that of wildlife [149]. Additionally, the ready availability of oocytes is a primary reason why domestic taxa have been used as oocyte donors and recipient surrogate mothers in conservation-related efforts [150].

Our investigations of published literature and interviews of cloning practitioners (Table 3) suggest that the leading reasons why cloning projects were not performed at larger scales or were terminated were due to funding and resource constraints rather than biological barriers inherent to the methods used nor insufficient knowledge of husbandry and reproductive biology. Regulatory restrictions, opposition or disinterest from certain stakeholder groups, and lack of political support were also reported as major factors influencing the parameters of cloning efforts (Table 3). For two efforts (banteng and sturgeon), the reason for termination was noted that other reproductive technologies were applicable or more promising to pursue/apply. However, in general, scientific difficulties in the cloning process were not cited as the basis for terminating efforts. Importantly, we found no reports that cited animal welfare as a problematic barrier to the conservation cloning efforts in question. If we accept claims from published interviews, cloning efforts have tended to exceed the welfare conditions required in animal research [108], a central principle for all animal programs within the Association of Zoos and Aquariums [151] within which multiple cloning research efforts have been pursued.

To date, the ephemeral nature of conservation-cloning efforts has not been examined in published reviews. We hypothesize that problematic narratives of cloning may contribute to the lack of continuing cloning efforts. Although there is no evidence in previous literature, we hypothesize that the high-profile nature of disappointments (i.e., early deaths, sterility) in early conservation cloning efforts came to dominate both public and scientific dialogues, overshadowing their value to scientific progress, and even overshadowing successful contemporary cloning achievements. While wildlife cloning successes have been inconsistently reported in previous reviews, the failures of the guar and bucardo have been universally cited [29,30,31,32,33,34,36,150]. We speculate that greater attention to failures in wildlife cloning, which receive more public interest than domestic taxa in popular media, has inflated commonly believed misconceptions that cloning cannot yield healthy offspring [152]. There is a lack of research of how social perceptions may have affected progress, or lack thereof, for applying cloning within conservation management. If there is a general impression that conservation cloning is not viable, which is interpretable from the content of most reviews published in peer-reviewed journals, then further research could be discouraged and limit funding interests within an already underfunded field. It has been shown that public use of particular science research outputs affects available funding [153]. Public “use” can be seen as equivalent to public support for a particular field of science. Higher public use of science has been shown to drive higher funding [153]. An increased understanding regarding why cloning fails, technical improvements in methodology, non-scientific factors shaping past efforts, and awareness of cloning successes, as we have cursorily reviewed here, each deserve individual and deeper analytical review. Such attention will aid in advancing research as well as narratives of cloning.

Other factors contribute to the lack of consideration for cloning applications by the conservation community. The incomplete narratives of the state of cloning technology stem from the fact that a large portion of data and methodology is either absent or has not been published in peer-review, and thus has not been vetted by the scientific community. However, in terms of long-term studies of clone health and longevity, funding limitations are once again a primary constraint. For many cloned taxa, life expectancy of natural born individuals in human care facilities can exceed 10 years, with some taxa with life expectancies reaching half a century (Table 3). Therefore, inter-generational research of these taxa requires multiple decades and can not only surpass the lifespan of the careers of principal investigators, but also requires decades of sustained animal care funding.

A multitude of factors, all of which stem from limited funding, have contributed to the disjointed progression of research and applications for conservation cloning compared to non-conservation research and commercial enterprises. We are suggesting that by disregarding the sporadic nature of conservation cloning research across a range of taxa and the limited experimental data that have been obtained in comparison to model systems such as the mouse, that it is premature to dismiss cloning as a conservation option worthy of more experimental evaluation. With renewed, adequately funded efforts performed at greater scales, testing new variations of oocyte/recipient taxa, and employing knowledge gained from advancements in domestic taxa [154], it is more probable that healthy fertile adult clones can be produced for many of the taxa for which previous efforts failed.

### 3.3. Considerations for Applied Conservation Cloning in Practice

Cloning offers a number of ways to enhance the management of *ex situ* populations using both frozen zoos and wild populations as sources of genetic material [2,29,30,34,35,36,155,156,157]. Though several other advanced reproductive technologies have been used in conservation, cloning is currently the only means of reproducing individual organisms from somatic cells for a diversity of taxa (producing offspring from *in vitro* gametogenesis using induced pluripotent stem cells has so far only been achieved in mice, [158,159]). Although cloning is not a universally applicable technology, conceivably it could be used to reproduce individuals from mammalian, fish, and amphibian taxa for conservation purposes.

The establishment and genetic management of *ex situ* populations of threatened taxa is an area of conservation in which cloning could be applied now. Maintaining *ex situ* populations of wild taxa is increasingly vital to safeguarding the security of healthy populations in the wild, as noted in the OnePlan approach of IUCN [160] and the One Conservation Proposal [154]. The devastating 2019–2020 fires in Australia are a key example of why *ex situ* populations are a lifeline in the face of human driven climate change. As the fires and the ash-laden floods that ensued created mass mortality events for entire ecosystems, conservationists rushed to capture and save individuals in the wild and bring them to safe harbor *ex situ* [161,162,163]. With populations of many taxa declining over the world [164,165], it is likely that *ex situ* populations, and therefore strategic reproductive strategies including cloning, will be critical for long-term recovery of many taxa [166].

For instance, cloning could be used to establish founding *ex situ* breeding populations of threatened taxa without removing individuals from the wild, if cell cultures can be established from non-lethal tissue biopsies. This is a particularly attractive application for rare fish and amphibian taxa which develop instinctively without the need for parental nurturing. Cloning has already been proven by historic efforts for several currently threatened amphibians (Table 1). Though no healthy adult amphibian clones have been produced from the donor cells of metamorphosed adults, cells of tadpoles have yielded healthy clones. Tail clippings from tadpoles could be collected non-lethally. Another limitation to duplicating wild amphibian populations is that cloning will need to utilize primary tissue samples without cell culture expansion, as has been done in almost all previous amphibian cloning studies. This is because to date the only tissue samples that have consistently yielded cell cultures have been from internal organs collected post-mortem [167], which effectively limits options for collecting tissues from wild individuals. Although there are limitations to cell culture and biobanking, this is an area highly worth pursuing research and applications given the current biodiversity crisis. It is estimated that nearly 41% of known amphibian taxa are at risk of extinction [168,169]. *Ex situ* populations of amphibians are becoming increasingly important as losses of populations and entire taxa due to the rapid spread of chytridiomycosis occurs. As species rely more and more on *ex situ* populations for recovery, the value of advancing biobanking and reproductive technologies becomes more apparent in terms of overall program costs. A recently published model by Howell et al. [170] for the threatened Oregon Spotted Frog, *Rana pretiosa*, estimated that back-crossing females every generation (at 3-year intervals) with cryopreserved founder sperm would allow a breeding program to retain 90% founder heterozygosity over 100 years with 97% fewer breeding individuals (58 versus 1826) and 96% lower cost over the cumulative century (USD 21 million versus USD 537 million). The ability to preserve and restore female genomes for breeding through cloning would further reduce resources and costs to achieve genetic management goals [171].

Establishing *ex situ* populations for threatened taxa before they become critically endangered improves the prognosis of recovering taxa in two ways [154]. Selecting founders when populations are still large captures a greater extent of standing genetic variation than can be expected when populations reach the nadir of a bottleneck. Secondly, working to build animal care knowledge, which is greatly enhanced with adequate time and numbers, reduces potential conflicts between removing animals from the wild and establishing robust populations under human care. When only a few individuals of a taxa remain, every loss is significant and action is urgent. With established cell cultures, cryopreservation, and cloning techniques for a specific taxon, losses are potentially recoverable and actionable timeframes potentially indefinite [2,29,157].

The reality of maintaining *ex situ* populations is that available capacity constrains population sizes. Populations that persist in small numbers over multiple generations inevitably lose genetic variation due to genetic drift. Cloning offers the ability to mitigate drift and restore or introduce new variation periodically from wild populations or biobanked cells. Variation is the underlying foundation of adaptation, hence maintaining the highest genetic diversity possible in *ex situ* populations is key to breeding individuals capable of adapting and surviving when introduced to the wild [172]. As noted earlier with the case example of amphibians, biobanking and advanced reproductive technologies can greatly reduce the numbers of individuals needed to maintain genetic diversity.

For populations that have already lost significant genetic variation during historic bottlenecks, cloning from cryopreserved resources offers a means to genetic rescue that would otherwise not be possible. Genetic rescue conventionally has been the practice of introducing new, healthy and unrelated individuals into genetically impoverished populations exhibiting or presumed to be experiencing inbreeding depression. The underlying hypothesis is that increasing genetic variation and associated heterozygosity in a population increases the population’s viability, as measured by positive population growth [173], which has been the result for ~95% of genetic rescue interventions [174]. As with establishing *ex situ* populations, we advocate for early, proactive intervention. Like most conservation practice, genetic rescue has been a reactive discipline, implemented only when populations are in clear crisis. There is an emerging paradigm shift to perform genetic rescue proactively, as a form of managing gene flow, before populations exhibit declines and deleterious inbreeding phenotypes [175,176,177]. The effectiveness of using genetic rescue early and periodically has been shown in genetic rescue simulations for two taxa that have been models for the benefits of genetic rescue and managed gene flow historically [173], the greater prairie chicken (*Tympanuchus cupido pinnatus*) and the Florida panther (*Puma concolor coryi*). [178,179]

While previously published reviews have consistently asserted that cloning for conservation requires further empirical research before subsequent application [33,34,36,180], the two are not mutually exclusive. Through the pursuit of cloning for applied conservation, scientists and practitioners produce valuable data for assisted reproduction. Conservation cloning can proceed with the intention to move beyond proof-of-principle research toward conservation action, consistent with the IUCN OnePlan approach. If programs secure adequate funding and integrate the following guidelines, conservation cloning efforts can fill targeted knowledge gaps and optimize taxa-specific cloning protocols while working to reproduce individuals of strategic value to management and recovery.

Cloning research should be advanced using taxa for which cloning can provide benefits to conservation and management. Individuals to be cloned should be selected for their strategic value for long-term management of taxa-specific genetic variation for many generations into the future. Conservation cloning efforts would ideally meet the overlapping criteria shown in Figure 2. If existing knowledge is not extensive, but the value of cloning for conservation is significant and urgent, research would preferably be designed to fill such knowledge gaps as cloning trials are performed.Biopsies to obtain cells from living donors should be minimally invasive, and in the case of *ex situ* animals should be obtained opportunistically during routine medical exams or other procedures to minimize handling that could induce stress. Donor cells should be cultured, expanded, and cryopreserved to save cells for future conservation purposes. If possible, cells should be reprogrammed to yield induced pluripotent stem cells, establishing a nearly inexhaustible resource [181].Oocytes and recipient surrogate mothers of common, non-threatened taxa with well-understood reproductive biology and husbandry should receive highest consideration. Not only does this practice alleviate welfare concerns for the focal taxa, but provides the ability to expand populations independent of natural breeding seasons, as domestic taxa can be bred continuously [154].The number of recipients and embryos transferred to establish pregnancies for mammalian taxa should be performed at scales in which live births can be expected. Commercial cloning companies and private ranchers can provide valuable knowledge and capacity to achieve success. For example, private ranches can be of significant value for cloning wild ungulates. Private ranches possess significantly larger herds of numerous exotic ungulates than zoos [182].Collection of data to build comprehensive baselines of normal and pathological embryonic, prenatal, perinatal, and postnatal development should be undertaken, incorporating control studies to enable research to identify developmental abnormalities in the cloning process.The results of cloning efforts need to be reported through peer-reviewed, open source outlets. If authors can show genetic evidence to support cloning claims, then journals need to allow the publishing of observations of cloned individuals despite the proprietary methods that may be employed to produce cloned embryos.Observations and analysis of health, behavior, and longevity should be made and reported throughout the lifetime of clones and at least one generation of offspring.

With appropriate levels of experimental rigor, conservation cloning efforts are more likely to be fruitful and trustworthy. To achieve this, cloning interventions will require sustained funding, and political and stakeholder support for durations of years and potentially decades. The program to produce fertile and reproductive black-footed ferret clones has been ongoing for over a decade since its inception, over half of which involved recovery team engagement, program design, fundraising, and iterative science before actual cloning began.

The programs cloning black-footed ferrets and Przewalski’s horses were used as case examples to develop the recommendations listed above. These cloning efforts are the first genetic rescue cloning efforts in history, operating under the “proactive” genetic rescue paradigm to increase genetic diversity before inbreeding yields significant fitness problems. The aim of these initiatives is to clone historic individuals with underrepresented or entirely absent genetic variation in present day generations to increase standing genetic diversity. While reintroduced wild populations of Przewalski’s horses and black-footed ferrets are capable of population expansion, the species have still experienced considerable losses. The severe bottlenecks these taxa experienced (black-footed ferret has 7 founders, Przewalski’s horse has 12 wild Przewalski’s horse founders) have severely eroded their genetic diversity [183,184], complicating their long-term management *ex situ* and creating concerns for their long-term adaptability in changing wild environments. Due to habitat fragmentation, gene-flow among reintroduced wild populations is also virtually impossible without human intervention, creating the conditions under which inbreeding depression may occur cryptically even though populations exhibit positive growth, as has been documented in reintroduced populations [185]. Even if wild populations become relatively large, reductions in genetic diversity in their past may diminish fitness over time. The black-footed ferret and Przewalski’s horses are not rare cases. Hundreds of animal species have undergone severe bottlenecks and are currently at varying stages of population recovery [165,186]. in which biobanking and reproductive technologies could provide the same genetic rescue options in the future.

However, species are not saved by technological advancements, but by the dedication and devotion of diverse conservationists, spanning participants from local communities to internationally collaborative teams. Conservation is a discipline built upon cultural values, with the recognition that a biodiverse world is not only integral to enriching the human experience, but that wild taxa have innate value. As responsible stewards, conservationists must carefully consider all aspects of interventions when saving a taxon, from impacts on ecosystems to the welfare of individual organisms. In this regard, new advanced reproductive technology initiatives are also working to lead by example, as members of the black-footed ferret and the northern white rhinoceros genetic rescue initiatives have worked to analyze and develop broadly applicable ethical frameworks for considering the use of cloning and advanced reproductive technologies in conservation [187,188].

It is our hope that the new genetic rescue cloning initiatives of the black-footed ferret and Przewalski’s horse inspire renewed interest and usher in a never-before-seen era of practical cloning applications in conservation. With the immense declines of wildlife populations, the core of the biodiversity crisis is not only the extinctions of taxa, but the extinction of adaptive genetic variation that leaves fragmented populations vulnerable to future changes, even as their populations recover in numbers [164,165,166,186]. Collections of cryopreserved viable cells and conservation breeding programs are no longer insurance policies in the battle to conserve biodiversity, they are needed at the front lines.

## 4. Conclusions

A thorough evaluation of the history of cloning, presented here, reveals that at least one, and oftentimes numerous healthy reproductive clones have been produced for the majority of the 56 total taxa that have reported live births/hatching of cloned individuals. The reliability and reproducibility of cloning model organisms and domestic taxa has led to commercially profitable cloning services for animal agriculture, sport, and companionship industries over the past 25 years. While conservation-relevant cloning research programs have been smaller scale and intermittent compared to the routine cloning activities of biomedical research and commercial industries, there have been many diverse wild taxa successfully cloned for the purpose of building towards meaningful conservation applications. A total of 14 taxa have been successfully cloned for conservation research (and most recently the first true applications). Collectively these efforts have produced 72 live births overall for conservation research. Of those 14 species, 38 clones (53%) of 10 taxa (71%) survived to the weaning stages with reports of fertility for 6 of those species. Many wildlife clones met or exceeded their life expectancies and all that reached adulthood were reported as healthy and normal developmentally. Although the number of wildlife clones pales in comparison to the thousands of cloned animals produced for commercial and biomedical research purposes, the details of fertility, health, and longevity of wildlife clones serve to dispel widely held misconceptions. Our inquiries to scientists that led previous conservation relevant cloning research revealed that the smaller scales and intermittent nature of wildlife cloning was shaped by many factors, prominently limited funding. With greater funding investment in this field cloning can be strategically applied to achieve conservation outcomes that no other available method may deliver, as exemplified by the Przewalski’s horse and black-footed ferret programs.

## Figures and Tables

**Figure 1 animals-15-00989-f001:**
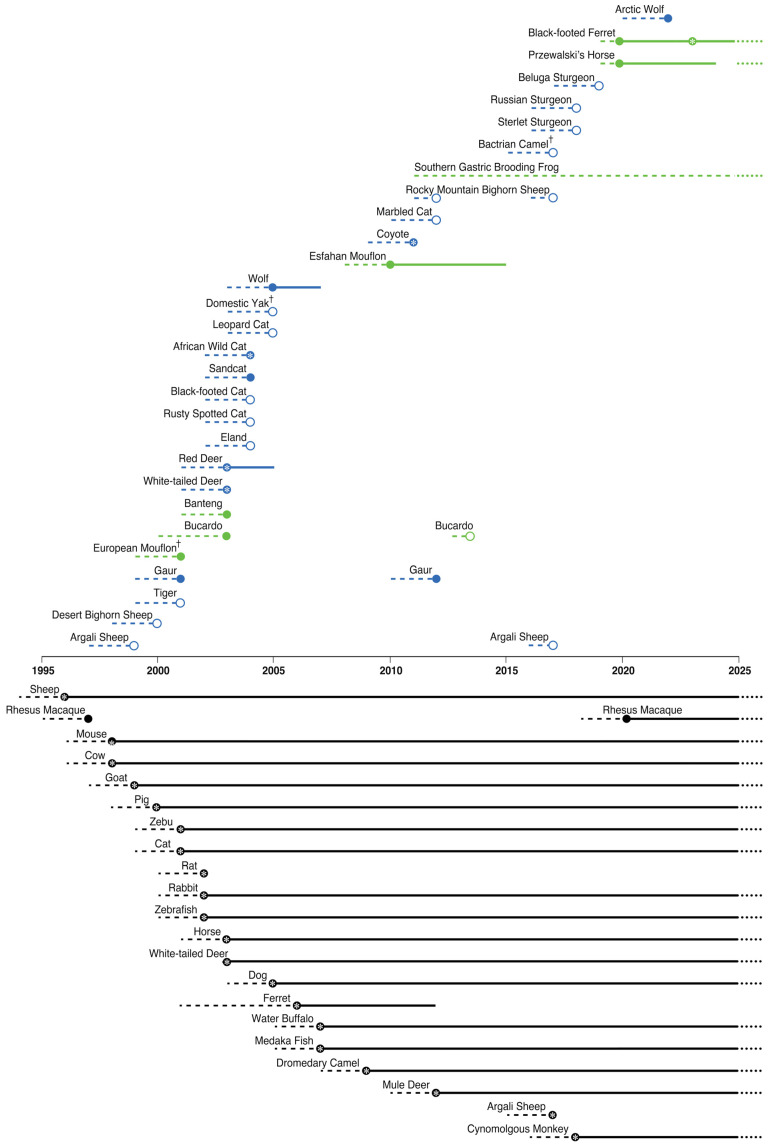
Duration of adult somatic cell cloning efforts by taxon and purpose. The purposes of each cloning effort are indicated by color and position: conservation-relevant cloning efforts are above the timeline with application-oriented efforts in green and research-oriented efforts in blue, while commercial or model research cloning fall below the timeline all colored black. Taxa marked with † are domestic taxa cloned as proxies for conservation-oriented research, but in the case of the European mouflon it was mistakenly classified as a wild species at the time. Two wild taxa that were the focus of conservation cloning research were also cloned for commercial purposes: the White-tailed Deer and Argali Sheep, and are therefore listed both above and below the timeline. Dashed lines represent cloning research and development preceding live births. Two years was estimated as the average research and development lead-up time for taxa, unless otherwise stated in publications or personal communication. Empty circles indicate efforts that failed to produce live births. Solid circles indicate the first achieved live births. Circles with stars indicate that fertile adult clones were produced. Solid lines indicate active cloning efforts for a given taxa, while breaks in lines represent inactivity. Beyond 2025, dotted lines indicate taxa for which it is likely that cloning activities will continue.

**Figure 2 animals-15-00989-f002:**
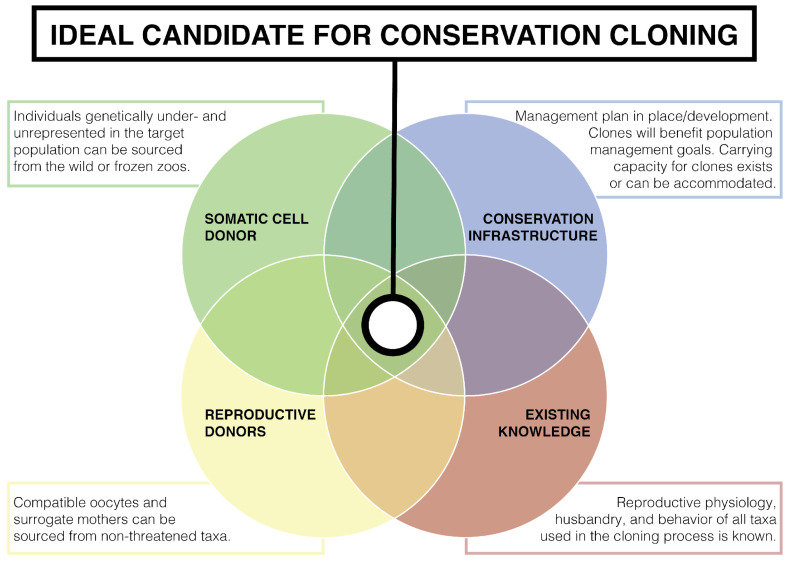
Ideal conservation cloning program Venn diagram.

**Table 1 animals-15-00989-t001:** Successfully cloned species and subspecies, ordered by year of first achieved live birth or hatching.

Common Name	Species Name	Year of First Live Birth/Hatching	Higher Classification	Wild or Domestic	Current IUCN Status	Same-Species Cloning	Cross-Species Cloning	Healthy Adult Clone(s) Produced	Evaluation of Reproductive Capability	Reference(s)	Nature of Reference
Dark-spotted frog	*Pelophylax nigromaculatus (formerly P. nigromaculata)*	1957	Amphibian	Wild	Near Threatened (assessed 2004); species was cloned before IUCN red list was formed		×	Yes	Fertile	[39]	Peer-reviewed publication.
Northern leopard frog	*Lithobates pipiens (formerly Rana pipiens)*	1957	Amphibian	Wild	Least Concern	×		Yes	Fertile	[38,42]	Peer-reviewed publication.
African clawed frog	*Xenopus laevis laevis*	1960	Amphibian	Domestic	Least Concern	×	×	Yes	Fertile	[43]	Peer-reviewed publication.
Mwanza frog	*Xenopus victorianus*	1961	Amphibian	Wild	Least Concern		×	Yes	Fertile	[44]	Peer-reviewed publication.
Durama pond frog	*Pelophylax porosus (formerly P. p. brevipoda)*	1963	Amphibian	Wild	Least Concern	×	×	Yes	Fertile	[45,46,47]	Peer-reviewed publication.
Japanese brown frog	*Rana japonica*	1963	Amphibian	Wild	Least Concern		×	Yes	Fertile	[45]	Peer-reviewed publication.
Montane brown frog	*Rana ornativentris (formerly R. ornaventris)*	1963	Amphibian	Wild	Least Concern		×	Yes	Unknown (Original reference could not be obtained, information gleaned from other papers)	[46,48]	Peer-reviewed publication.
Goldfish	*Carassisus auratus auratus*	1963	Fish	Domestic	Least Concern		×	Yes	Fertile	[49]	Peer-reviewed publication.
Bitterling	*Rhodeus sinensis*	1963	Fish	Wild	Least Concern	×	× (cross-genus)	Yes	Fertile	[49]	Peer-reviewed publication.
Common Asian carp	*Cyprinus carpio*	1963	Fish	Domestic	Wild populations Vulnerable (assessed 1996)	×	× (cross-species and cross-genus)	Yes	Fertile	[49]	Peer-reviewed publication.
Axolotl	*Ambystoma mexicanum*	1965	Amphibian	Domestic	Wild populations Critically Endangered (assessed 1986)	×		Yes	Fertile	[50]	Peer-reviewed publication.
Peter’s platanna	*Xenopus petersii*	1966	Amphibian	Wild	Least Concern		×	Yes	Unknown (Original reference could not be obtained, information gleaned from other papers)	[46]	Peer-reviewed publication.
Iberian ribbed newt	*Pleurodeles waltl*	1970	Amphibian	Wild	Near Threatened (assessed 2006)	×	×	Yes	Fertile	[51]	Peer-reviewed publication.
Edough ribbed newt	*Pleurodeles poireti*	1970	Amphibian	Wild	Endangered (assessed 2004)	×	×	Yes	Unknown (Original reference could not be obtained, information gleaned from other papers)	[51]	Peer-reviewed publication.
Gold-spotted pond frog	*Pelophylax plancyi (formerly P. chosenicus)*	1972	Amphibian	Wild	Vulnerable (assessed 2004)		×	Yes	Fertile	[46]	Peer-reviewed publication.
Common frog	*Rana temporaria*	1972	Amphibian	Wild	Least Concern		×	Yes	Infertile	[48]	Peer-reviewed publication.
Edible frog	*Pelophylax esculenta (possibly. P. ridibundus or P. lessonae)*	1972	Amphibian	Wild	Not classified, considered common		×	No	NA	[47]	Peer-reviewed publication.
Wild goldfish	*Carassius auratus*	1980	Fish	Wild	Least Concern	×	× (cross-subspecies)	Yes	Fertile	[52]	Peer-reviewed publication.
Mouse	*Mus musculus*	1981	Mammal	Domestic	Least Concern	×		Yes	Fertile	[53]	Peer-reviewed publication.
Grass Carp	*Ctenopharyngoden idellus*	1984	Fish	Wild	Not classified, considered common		× (cross-genus)	Yes	Fertile	[54]	Peer-reviewed publication.
Sheep	*Ovis aries*	1986	Mammal	Domestic	Not Applicable	×		Yes	Fertile	[55]	Peer-reviewed publication.
Cattle	*Bos taurus*	1987	Mammal	Domestic	Not classified, not applicable	×		Yes	Fertile	[56]	Peer-reviewed publication.
Rabbit	*Oryctolagus cuniculus domestica*	1988	Mammal	Domestic	Not classified, not applicable	×		Yes	Fertile	[57]	Peer-reviewed publication.
Pig	*Sus scrofa domesticus*	1989	Mammal	Domestic	Not classified, not applicable	×		Yes	Fertile	[58]	Peer-reviewed publication.
Loach	*Paramisgurnus dabryanus*	1990	Fish	Wild	Not classified, considered common	×		Yes	Fertile	[59]	Peer-reviewed publication.
Goat	*Capra hircus*	1991	Mammal	Domestic	Not Applicable	×		Yes	Fertile	[60]	Peer-reviewed publication.
Rhesus macaque	*Macaca mulatta*	1997	Mammal	Wild	Least Concern	×		Clones born in 2020 are still alive and nearing age of sexual maturity.	NA	[61,62,63]	Peer reviewed publications, current status of clones provided by Dr. Qiang Sun.
Medaka	*Oryzias latipes*	1999	Fish	Domestic	Least Concern	×		Yes	Fertile	[64,65]	Peer-reviewed publication.
Zebu	*Bos indicus*	1999	Mammal	Domestic	Least Concern		×	Yes	Fertile	[66,67,68]	Peer-reviewed publications.
Gaur	*Bos gaurus*	2001	Mammal	Wild	Vulnerable		×	No	NA	[69,70]	Peer-reviewed publication.
European mouflon	*Ovis aries musimon*	2001	Mammal	Feral Domestic	Not classified, not applicable (was considered a wild species at time of cloning, *O. gmelini* musimon, which was classified as Near Threatened)		×	Yes	Reached weaning age, but not sexual maturity	[71]	Peer-reviewed publication. Fertility information provided by Dr. Pasqualino Loi.
Domestic Cat	*Felis catus*	2001	Mammal	Domestic	Not classified, not applicable	×		Yes	Fertile	[72]	Peer-reviewed publication.
Zebrafish	*Danio rerio*	2002	Fish	Domestic	Least Concern	×		Yes	Fertile	[73,74]	Peer-reviewed publication.
Rat	*Rattus norvegicus domestica*	2002	Mammal	Domestic	Not classified, not applicable	×		Yes	Fertile	[75]	Peer-reviewed publication.
Bucardo (died)	*Capra pyrenaica pyrenaica*	2003	Mammal	Wild	Extinct (nominate species Least Concern)		×	No	NA	[76]	Peer-reviewed publication.
Banteng	*Bos javanicus*	2003	Mammal	Wild	Endangered		×	Yes	Infertile	[77]	Conference abstract. Fertility reported here for the first time.
Horse	*Equus caballus*	2003	Mammal	Domestic	Not classified, not applicable	×		Yes	Fertile	[78]	Peer-reviewed publication.
White-tailed deer	*Ococoileus virginianus*	2003	Mammal	Wild	Least Concern	×		Yes	Fertile	[79,80]	All information reported in popular press.
Red deer	*Cervus elaphus*	2003	Mammal	Wild	Least Concern	×		Yes	Fertile	[81]	Peer-reviewed publication. Fertility information provided by Dr. Debbie Berg.
Fruitfly	*Drosophila melanogaster*	2004	Insect	Domestic	Not classified, not applicable	×		Yes	Fertile	[82]	Peer-reviewed publication.
African wild cat	*Felis sylvestris lybica*	2004	Mammal	Wild	Least Concern		×	Yes	Fertile	[83]	Peer-reviewed publication.
Sand cat *	*Felis margarita*	2004	Mammal	Wild	Least Concern (Near Threatened at time of cloning)		×	No	NA	[84]	Peer-reviewed publication.
Domestic Dog	*Canis lupus familiaris*	2005	Mammal	Domestic	Not classified, not applicable	×		Yes	Fertile	[85]	Peer-reviewed publication.
Gray Wolf	*Canis lupus lupus*	2005	Mammal	Wild	Least Concern		× (cross-subspecies)	Yes	Unknown	[86,87]	Peer-reviewed publication.
Domestic Ferret	*Mustela putorius furo*	2006	Mammal	Domestic	Least Concern	×		Yes	Fertile	[88]	Peer-reviewed publication. Fertility information provided by Dr. John Engelhardt.
Water Buffalo	*Bubalus bubalis*	2007	Mammal	Domestic	Not classified, not applicable	×		Yes	Fertile	[89,90]	Peer-reviewed publication
Dromedary Camel	*Camelus dromadarius*	2009	Mammal	Domestic	Not classified, not applicable	×		Yes	Fertile	[91,92]	Peer-reviewed publication.
Coyote	*Canis latrans*	2011	Mammal	Wild	Least Concern		×	Yes	Fertile	[93]	Peer-reviewed publication. Fertility information provided by Dr. Hwang Suk.
Mule deer	*Ococoileus hemionus*	2012	Mammal	Wild	Least Concern	×		Yes	Fertile	[94]	Reported in popular press. Fertility information provided by Shawn Walker.
Esfahan mouflon	*Ovis gmelini isphahanica*	2015	Mammal	Wild	Near Threatened		×	Unknown	Unknown	[95,96]	Live births reported in peer-reviewed publication in 2010, single live birth reported by researchers to press in 2015.
Bactrian camel	*Camelus bactrianus*	2017	Mammal	Domestic	Not classified, not applicable		×	No	NA	[97]	Peer-reviewed publication.
Argali	*Ovis ammon*	2017	Mammal	Wild	Near Threatened		×	Yes	Fertile	[98,99]	Reported in popular press and government press release from court case documents.
Cynomolgus monkeys	*Macaca fascicularis*	2018	Mammal	Wild	Endangered (assessment published 2020); species was listed as Least Concern at the time of cloning	×		Yes	Fertile	[100,101]	Peer reviewed publication. Current status of clones provided by Dr. Qiang Sun
Przewalksi’s Horse	*Equus przewalskii*	2020	Mammal	Wild	Endangered		×	Clone born in 2020 is alive and expected to reach sexual maturity in 2025; second clone will reach sexual maturity in 2028	Cannot be evaluated yet	[102]	Peer-reviewed publication.
Black-footed ferret	*Mustela nigripes*	2020	Mammal	Wild	Endangered		×	Yes	Fertile	[3,4]	Preprint and government press release.
Arctic wolf	*Canis lupus arctos*	2022	Mammal	Wild	Least Concern		×	Unknown	Not yet reported	[103]	Press release.

The designation × denotes which method of cloning has been applied successfully to the species. When recording reproductive capability the designation NA (not applicable) is provided for species in which no individual clone reached sexual maturity, and therefore reproductive capability could not be ascertained. * The sand cat is not considered endangered today, and although listed as Near Threatened from 2002 to 2016, it was likely not endangered. Its downgraded listing from Near Threatened to Least Concern was not due to recovery, but due to a reassessment. The species is rarely seen, but has been consistently observed [104]. The Mediterranean population, however, is still listed as Near Threatened.

**Table 2 animals-15-00989-t002:** Detailed information regarding conservation related cloning efforts.

**Common Name**	Species Name	Status	Method	Year(s)	Oocyte Donor	Somatic/Oocyte Donor Evolutionary Divergence Estimated (Range) MYA *	Recipient Surrogate Mothers	Number of Embryos Transferred/Fostered for Development	Number of Live Births/Hatching	Percentage Implanted/Fostered Embryos Resulting in Live Births/Hatching	Number of Clones Reaching Weaning Age	Reproductive Capability	References	Nature of References
Argali	*Ovis ammon*	Near Threatened	Cross-species	1999, 2017	*Ovis aries* (domestic)*, O. c. canadensis* × *O. aries hybrid*	0.96 (0.4730–1.4406)	*Ovis aries* (domestic)*, O. c. canadensis x O. aries hybrid, O. c. canadensis x O. aries/O. gmelini (Rocky Mountain Bighorn x Texas Dall Sheep hybrid)*	28, 165	0, 1	0, 0.6%	0, 1	Cloned argali was used to produce >50 offspring via hybridization with Rocky Mountain Bighorn Sheep, *Ovis canadensis canadensis*	[98,99,105]	Peer reviewed publication, court case documents
Desert Bighorn Sheep	*Ovis canadensis nelsonii (formerly O. c. mexicana)*	Least Concern	Cross-species	2000, 2017	*Ovis aries* (domestic)	3.14 (2.30–4.97)	*Ovis aries* (domestic)	223	0	0	0	NA	[106,107]	Conference abstract, book series, and information provided by Dr. Mike Kjelland.
Gaur	*Bos gaurus*	Vulnerable	Cross-species	2001, 2012	*Bost taurus*	3.76 (2.86–5.42)	*Bos taurus*	44 **	1, 1	2.4% (averaged); 0–11% per cell line/treatment	0	NA	[69,70,108]	Peer reviewed publication reports of viable cloned pregnancies in 2000, live birth reported to press by San Diego Zoo Global, Advanced Cell Technologies, and TransOva Genetics. Live births from another research group were Fertile in peer-reviewed literature in 2012.
Tiger	*Panthera tigris altaica*	Endangered	Cross-species	2001	Could not obtain original reference.	-	-	-	0	0	0	NA	[109]	Conference abstract is referenced in other papers, but original reference could not be obtained to extract data regarding each step of the cloning process.
Banteng	*Bos javanicus*	Endangered	Cross-species	2003	*Bos taurus*	3.76 (2.86–5.42)	*Bos taurus*	NA	2	NA	1	Infertile	[108]	Abstract from conference. Reproductive capability and Reason for program termination presented in this publication by co-authors of previous peer-reviewed report.
Bucardo	*Capra pyrenaica pyrenaica*	Subspecies Extinct (Evolutionarily Torpid), Nominate species Least Concern	Cross-species	2003, 2013	*Capra hircus*	2.85 (1.25–5.61)	*Capra pyrainaica hispanica and Capra pyrenaica hispanica x Capra hircus hybrids*	154, NA	1, 0	0.6% (total)	0	NA	[76]	Peer-reviewed publication. The research team renewed efforts in 2013, but produced no pregnancies. Personal communication Dr. Alberto Fernándaz-Arias.
Red deer	*Cervus elaphus*	Least Concern	Same-species	2003–2005	*Cervus elaphus*	-	*Cervus elaphus*	84	10	12% (averaged)	8	Clones produced normal sperm, were never bred.	[81]	Peer-reviewed publication, personal communication Debbie Berg.
White-tailed Deer	*Odocoileus virginianis*	Least Concern	Same-species	2003 onward	*Odocoileus virginianis*	-	*Odocoileus virginianis*	NA	NA	NA	1	Cloned deer was reported to produce offspring, with up to three generations of descendants.	[79,80]	First clone Fertile by Texas A&M University and ViaGen Pets & Equine, but never published in peer-review. Total number of adult clones produced provided by ViaGen Pet’s and Equine. Personal communication Shawn Walker.
Eland	*Tragelaphus oryx*	Least Concern	Cross-species	2003	*Bos taurus*	17.3 (14.3–19.3)	-	-	0	0	0	NA	[110]	Conference abstract is referenced in other papers, but original reference could not be obtained to extract data regarding each step of the cloning process.
Sandcat	*Felis margarita*	Least Concern	Cross-species	2004	*Felis catus*	3.6 (1.13–5.10)	*Felis catus*	1600	14	0.3–1.8% per treatment/cell line	0	NA	[84,111]	Peer-reviewed publications.
Black-footed cat	*Felis nigripes*	Vulnerable	Cross-species	2004	*Felis catus*	4.35 (3.36–5.92)	*Felis catus*	698	0	0	0	NA	[84,111]	Peer-reviewed publications.
Rusty spotted cat	*Prionailurus rubiginosus*	Near Threatened	Cross-species	2004	*Felis catus*	9 (5.6–11.9)	*Felis catus*	201	0	0	0	NA	[84,111]	Peer-reviewed publications.
African wildcat	*Felis sylvestris lybica*	Least Concern	Cross-species	2004	*Felis catus*	2.8 (1.40–5.10)	*Felis catus*	1627	17	1% (averaged)	8	Clones were bred, one generation of descendants published.	[111]	Peer-reviewed publication.
Domestic Yak ***	Bos grunniens	Not Listed, Not applicable	Cross-species	2005	*Bos taurus*	4.36 (2.72–6.46)	*Bos taurus*	324	0	0	0	NA	[112]	Peer-reviewed publication.
Gray Wolf	*Canis lupus lupus*	Least Concern	Cross-subspecies	2005, 2007	*Canis lupus familiaris*	(0.02–0.04)	*Canis lupus familiaris*	251, 372	2, 6	0.8%; 0–9.5% per oocyte donor/recipient combination	2,3	NA	[86,87,113]	Two independent research groups published successful results in peer-reviewed literature. Reports of intent to breed in popular press.
Leopard cat	Prionailurus bengalensis	Least Concern	Cross-species	2005	*Felis catus*	9 (5.6–11.9)	*Felis catus*	435	0	0	0	NA	[114]	Peer-reviewed publication.
Esfahan Mouflon	*Ovis gmelini isphahanica*	Near Threatened	Cross-species	2010–2015	*Ovis aries* (domestic)	1.17 (0.47–8.10)	*Ovis aries* (domestic)	12	2, NA	16% (averaged); NA	0, NA	Unknown	[95,96]	Two live births reported in peer-reviewed publication in 2010, single live birth reported by researchers to press in 2015.
Coyote	*Canis latrans*	Least Concern	Cross-species	2011	*Canis lupus familiaris*	3.04 (1.39–4.51)	*Canis lupus familiaris*	320	8	2.5% (averaged)	8	Clones were bred, two litters have been born to cloned parents to date.	[93]	Peer-reviewed publication. Reproduction information provided by Dr. Hwang Woo-Suk.
Rocky Mountain Bighorn Sheep	*Ovis canadensis canadensis*	Least Concern	Cross-species	2012	*Ovis aries*	3.14 (2.30–4.97)	*Ovis aries* (domestic)	NA	0	0	0	NA	NA	Information provided by Dr. Mike Kjelland.
Marbled cat	*Pardofelis marmorata*	Near Threatened	Cross-species	2012	*Felis catus*	15.2 (12.2–16.6)	*Felis catus*	461	0	0	0	NA	[115]	Peer-reviewed publication.
Southern Gastric Brooding Frog	*Rheobatrachus silus*	Extinct	Cross-species	2013-	*Mixophyes fasciolatus*	141 (123–149)	-	NA	0	0	0	NA	[116]	Popular press. Current status provided by Dr. Michael Archer
Bactrian Camel ***	*Camelus bactrianus*	Not Listed, Not applicable	Cross-species	2017	*Camelus dromedarius*	6.4 (2.4–9.5)	*Camelus dromedarius*	26	1	3.8% (total)	0	NA	[97]	Peer-reviewed publication.
Russian Sturgeon	*Acipenser gueldenstaedtii*	Critically Endangered	Cross-species	2018, 2019	*Acipenser ruthenus*	71 (8–71)	-	266	0	0	0	NA	[117,118]	Peer reviewed publications.
Sterlet	*Acipenser ruthenus*	Vulnerable	Same-species	2018	*Acipenser ruthenus*	-	-	129	0	0	0	NA	[117]	Peer-reviewed publication.
Beluga	*Huso huso*	Critically Endangered	Cross-species	2019	*Acipenser ruthenus*	158 (16–172)	-	130	0	0	0	NA	[117,118]	Peer reviewed publications.
Przewalksi’s Horse	*Equus przewalskii*	Endangered	Cross-species	2020, 2023	*Equus caballus*	(0.035–0.055)	*Equus caballus*	11	2	18% (total)	2	NA	[102]	Peer-reviewed publication..
Black-footed ferret	*Mustela nigripes*	Endangered	Cross-species	2020 onward	*Mustela putorius furo*	0.5	*Mustela putorius furo*	163	3	0–2.4%	3	One clone has produced 2 kits to date	[4,5]	Preprint and USFWS press release.
Arctic wolf	*Canis lupus arctos*	Least Concern	Cross-subspecies	2022	*Canis lupus familiaris*	(0.02–0.04)	*Canis lupus familiaris*	85	1	0.1%	1	NA	[103]	Press Release Announcement by SinoGene, China.

The number of embryos transferred, number of live births/hatching, and number of clones reaching weaning age are recorded per the years in which cloning attempts were made, with the data per year separated by a comma (,). When recording reproductive capability the designation NA (not applicable) is provided for species in which no individual clone reached sexual maturity, and therefore reproductive capability could not be ascertained. * Divergence times retrieved from [119] for all taxa except gray wolves [120] and Przewalski’s horse [121]. ** Percentage efficiency is calculated from 41 implanted embryos, as 3 fetuses were terminated early for scientific evaluation and may otherwise have resulted in viable clones. *** Although domestic species, the efforts to clone yak and bactrian camel were motivated as proof-of-concepts for their endangered wild counterparts.

**Table 3 animals-15-00989-t003:** Reasons for initiation and termination of conservation-related cloning efforts and accompanying information of longevity and causes of death.

Common Name	Species Name	Primary Purpose for Cloning	Reason for Program Termination	Longevity	Typical Lifespan	Causes of Death and Post-Natal Information.	References	Nature of Reference
Coyote	*Canis latrans*	Research: to advance conservation cloning science	Obtaining live birth satisfied end goal.	One lived 4 years, two lived 8 years, one lived 9 years. Four clones were still alive in 2020 at 9 years of age.	13–15 years	The first death was caused by wounds incurred from fighting, causes of two deaths unknown, one death due to heartworm parasitic infection. Clones were bred and produced six F1 offspring, one of which was cannibalized at less than one year of age, another died less than one year of age, cause unknown, and the remaining four were alive at six and four years of age in 2020.	Personal communication.	Information provided by Dr. Hwang Woo-Suk.
African wildcat	*Felis sylvestris lybica*	Research: to advance conservation cloning science	No response to inquiry; assumption that live births satisfied research end-goals	Three clones lived < 36 h, six lived < 30 days, one lived 1 month, one lived 3 years, two were terminated at 14 years. Four clones were still alive in 2020 at 16 years of age.	13–14 years	Two perinatal deaths due to bacterial infections. One perinatal death due to acute pneumonia from aspiration from bottle-feeding. Remaining perinatal deaths due to respiratory failure, potentially due to underdeveloped lungs. One clone died at 1 month of age due to poor fitness. One clone died at 3 years of age due to trauma. Two clones were euthanized due to age related issues at extreme old age. Four clones (50%) were alive and exceeding life expectancy in 2020. These clones produced eight F1 offspring, two of which died at one year of age (unknown reasons, suspected accident in enclosure), one euthanized at age 14 due to age related complications, and the remaining five were alive at 15 years of age in 2020.	[111]	Peer-reviewed publication, additional information provided by Lisa Murphy.
Gaur	*Bos gaurus*	Research: to advance conservation cloning science	Both studies achieved scientific end-goals; cloning the gaur by San Diego Zoo Global, Advanced Cell Technologies, and TransOva Genetics was an initial proof of concept for endangered bovids. Cloning additional gaur by San Diego Zoo Global was not pursued because it would not serve significant conservation value.	>48 h	Up to 26 years (typical range unknown)	The single clone died of dysentery.	[70,108]	Live birth and death in 2000 reported in press. Reason for termination reported from interviews in published book.
European mouflon	*Ovis aries musimon*	Research and Application: to advance conservation cloning science and recover deceased individuals within a small managed population. At the initiation of the work the species was classified as a distinct wild taxon, but now is considered a feral subspecies of domestic sheep.	Limited funding prevented further attempts. When initiated, Italy prohibited cloning of domestic animals unless for transgenic research, wild animals were exempted from the ban. Cloning mouflon presented an opportunity to advance cloning research as well as to recover the genotypes of deceased individuals within a small island population on Sicily.	14 months	8–12 years	The single clone was released to the same management area that the somatic donor lived. It died of pulmonary adenomatosis induced by infection. This was the same disease that caused the death of the somatic cell donor, and therefore susceptibility to the disease was not due to poorer fitness of the clone. Pulmonary infections are a major cause of multiple sheep taxa mortalities worldwide. Until onset of infection the clone was healthy and developed normally.	Personal communication.	Information provided by Dr. Pasqualino Loi.
Bucardo	*Capra pyrenaica pyrenaica*	Application: to recover a subspecies that had gone extinct from cryopreserved cell lines.	Limited funding and opposition from specific political and conservation stakeholders. A recent successful reintroduction of a related Ibex subspecies to the bucardo’s former range now precludes significant ecological value of recovering the extinct subspecies.	~7 minutes	16 years	The clone died due to complications arising from an extra, abnormal lung lobe. This was likely due to problematic early embryonic nuclear reprogramming or later gestational complications.	[76]	Peer-reviewed publication. Reason for program termination provided by Dr. Alberto Fernándaz-Arias.
Banteng	*Bos javanicus*	Application: genetic management of under-represented individual in *ex situ* zoo populations	Although cloning will be of value long-term, other lower cost genetic management options are currently adequate, negating justification to continue efforts to produce a fertile clone currently.	One clone lived 7 days, the second lived 7 years	Up to 27 years (typical range unknown)	The first clone was large and failed to gain strength, an abnormality in domestic cattle that occurs in higher frequencies in clones. The second individual was healthy throughout its life and died of injuries incurred in an accident with another animal in its enclosure, not uncommon for individuals in multi-individual enclosures.	[77]	Author Oliver Ryder, collaborator and co-author on previous peer-reviewed publications regarding this program, provided detailed information.
White-tailed Deer	*Odocoileus virginianis*	Research: first clone was produced for conservation cloning research; subsequent efforts for private enterprise (see Figure 1)	Conservation cloning research achieved its end-goal. Cloning of these taxa continues by commercial efforts.	First cloned deer lived 15 years, many individuals are currently alive.	6–14 years	The first cloned deer, Dewey, died of natural aging, having exceeded typical life expectancy. He was reported as healthy throughout his life.	[79,80]	Reported in popular press, additional information provided by Dr. Alice Blue-McClendon.
Red deer	*Cervus elaphus*	Research: unrelated to conservation	New Zealand Deer Industry stakeholders had no further interest in cloning deer.	Two lived < 1 week, one lived 5 years, one lived 7 years, two terminated at 8 years, three terminated at 10 years of age.	10–13 years	One calf died likely due to several observed pathologies (contracted flexor tendons in all legs, enlarged fatty liver, and incomplete lung inflation-these issues are observed in naturally reproduced individuals but expected at higher frequency in clones). One calf died due to parental neglect. One adult died from injury, breaking his neck on a fence, and another died of injuries incurred while fighting another clone while rutting-both causes of death not infrequent for males of all deer species. The remaining clones, born in 2003 and 2005 were euthanized together, at ages 8 and 10, due to lack of funding to continue animal care.	[81]	Peer-review publication, additional information provided by Dr. Debbie Berg.
Sandcat	*Felis margarita*	Research: to advance conservation cloning science	No response to inquiry; assumption that live births satisfied research end-goals	Nine clones lived < 36 h, three lived < 30 days, one lived 30 days, one lived 60 days.	Up to 13 years (typical range unknown)	Deaths were due to respiratory failure, possibly due to underdeveloped lungs. Four died of acute pneumonia resulting from aspiration during bottle-feeding.	[84]	Peer-reviewed publication.
Gray Wolf	*Canis lupus*	Research: to advance conservation cloning science	Obtaining live birth satisfied end goal.	4 years and 11 years	10–18 years	Exact causes of death for clones born in 2005 are unknown, assumed to be heat exposure and age related. Data for clones born in 2007 was not obtained.	[86,87,142]	Information only obtained for the first 2 cloned wolves produced by [86]. First death reported in popular press, additional information provided by Dr. Hwang Woo-Suk.
Esfahan Mouflon	*Ovis gmelini isphahanica*	Application: To increase population size of *ex situ* back up populations	No response to inquiry; assumption that live births satisfied research end-goals	<1 day, <14 days	8–12 years	Clones born in 2010 may have died due to organ abnormalities, potentially indicative of prematurity, though normal phenotypes of newborns for this species are unknown, so cause of death is inconclusive. The clone born in 2015 was reported healthy at 14 days of age. The current status of this clone could not be obtained.	[95,96]	Peer-reviewed publicaton and popular press.
Rocky Mountain Bighorn Sheep	*Ovis canadensis*	Research: to advance conservation cloning science	Limited funding.	NA	6–15 years	NA	Personal communication.	Information provided by Dr. Mike Kjelland.
Bactrian Camel	*Camelus bactrianus*	Research: proof-of-concept for conservation cloning of wild endangered Bactrian camels (*Camelus ferus*)	Wild Bactrian camel conservation groups were disinterested in pursuing conservation cloning; high demand for cloning dromedary camels for athletic and aesthetic competition has precluded further bactrian camel work.	<7 days	20–40 years	Clone died of acute septicemia.	[97]	Death reported in peer-reviewed publication. Reason for termination provided by Dr. Nisar Ahmed Wani.
Sterlet	*Acipenser ruthenus*	Research: to advance conservation cloning science	The primary scientist conducting cloning experiments performed this work to earn her doctorate degree. Once graduated, the program ended. Due to the low efficiency and difficulty of obtaining success in cloning, the laboratory group has shifted focus to other germ-cell based advanced reproductive techniques, including the more promising process of culturing, preserving, and transplanting of primordial germ cells to breeding surrogates.	NA	22–25 years	NA	[117]	Peer-reviewed publication. Reason for termination provided by Dr. Effrosyni Fatira and Dr. Martin Pšenička.
Russian Sturgeon	*Acipenser gueldenstaedtii*			NA	38 years	NA	[117,118]	
Beluga	*Huso huso*			NA	>100 years possible	NA	[117,118]	
Przewalksi’s Horse	*Equus przewalskii*	Application: genetic rescue of under-represented individual to increase adaptive diversity of *ex situ* and reintroduced *in situ* populations	NA—program is ongoing	Clones are alive at 1 and 4 years of age at time of manuscript submission	20–25 years	NA	[102]	Peer reviewed publication.
Black-footed ferret	*Mustela nigripes*	Application: genetic rescue to of an unrepresented individual (establishing a new founder) to increase adaptive diversity of *ex situ* and reintroduced *in situ* populations	NA—program is ongoing	Clones are alive at 1 and 4 years of age at time of manuscript submission	4–6 years	NA	[4,5]	Preprint and USFWS press release.
Arctic wolf	*Canis lupus arctos*	Research: to advance conservation cloning science	No response to inquiry; assumption that live births satisfied research end-goals	Clone is alive at over 2 years of age at time of submission	7-–10 years	NA	[103]	Press Release Announcement by SinoGene, China.

The lifespans (longevity) for clones of efforts that did not produce live births the information is entered as NA (not applicable). For clones that are still alive at the time of submission the causes of death is also entered as NA (not applicable).

**Table 4 animals-15-00989-t004:** Divergence times of fish and amphibian cross-species and cross-genus clones that yielded hatched offspring. * Divergence times retrieved from [119] for all taxa.

Higher Classicfication	Somatic Cell Donor	Oocyte Donor	Somatic/Oocyte Species Evolutionary Divergence in MYA *	Note	Reference(s)	Nature of Reference
Fish	*Rhodeus sinensis*	*Carassius auratus*	106 (68–102)	reciprocal transfers also produced live hatching	[49]	Peer-reviewed publication.
Fish	*Cyprinus carpio*	*Crassius auratus, Carassius carassius*	34 (21–46)	reciprocal transfers with C. auratus also produced live hatching, no reciprocal transfers with C. carassius performed	[49]	Peer-reviewed publication.
Fish	*Ctenopharyngoden idellus*	*Megalobrama amblycaephala*	12.1 (9.3–21.4)	reciprocal transfers also produced live hatching	[54]	Peer-reviewed publication.
Amphibian	*Pelophylax nigromaculata*	*Pelophylax porosus brevipoda*	9.92 (4.09–17.22)	reciprocal transfers also produced live hatching	[39]	Peer-reviewed publication.
Amphibian	*Xenopus victorianus*	*Xenopus laevis*	11.05 (7.89–14.40)	reciprocal transfers also produced live hatching	[44]	Peer-reviewed publication.
Amphibian	*Rana ornativentris*	*Rana japonica*	21.4 (13.7–24.4)	reciprocal transfers also produced live hatching	[46,48]	Peer-reviewed publication.
Amphibian	*Pleurodeles waltl*	*Pleurodeles poireti*	17.3 (10.1–19.3)	reciprocal transfers also produced live hatching	[51]	Peer-reviewed publication.
Amphibian	*Pelophylax esculenta*	*Pelophylax porosus brevipoda*	33 (21–42)	reciprocal transfers also produced live hatching	[46]	Peer-reviewed publication.
Amphibian	*Pelophylax chosenicus*	*Pelophylax porosus brevipoda*	9.92 (4.09–17.22)	reciprocal transfers also produced live hatching	[48]	Peer-reviewed publication.
Amphibian	*Rana temporaria*	*Rana japonica*	19.4 (11.4–24.4)	reciprocal transfers also produced live hatching	[47]	Peer-reviewed publication.

## Data Availability

All data is presented in this publication.

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
