# Peer review of "Towards Practical Conservation Cloning: Understanding the Dichotomy Between the Histories of Commercial and Conservation Cloning"

_animals, 2025, doi:10.3390/ani15070989_

Round 1

Reviewer 1 Report

Comments and Suggestions for Authors

1. Brief Summary

The paper explores the contrasting development paths and applications of cloning technologies for commercial purposes compared to conservation efforts. It highlights significant achievements in conservation cloning, such as the successful birth of endangered species like the black-footed ferret, challenging common misconceptions regarding the viability of cloning as a conservation tool. The paper emphasizes the potential conservation impact of cloning, with advances in understanding somatic cell cloning from the 1980s' frozen zoos to recent applications in genetic rescue.

2. General Concept Comments

Simply fantastic! Fascinating! Extraordinary! Phenomenal! This reviewer finds it difficult to find words to describe how rewarding it was to review this manuscript and shares a similar vision regarding cloning for conservation. The authors are to be commended for this incredible material. They have surpassed the horizon of speculation and false paradigms that exist in the field of species conservation concerning cloning. Not only have they compiled, but they have elucidated everything about cloning in conservation. This paper is set to break paradigms in species conservation, especially in the application of new technologies.

The manuscript provides a thorough historical analysis of cloning, underscoring pivotal developments and their implications for conservation biology. It offers critical insights by compiling a comprehensive list of successfully cloned taxa and presenting empirical data from diverse research outcomes.

This reviewer identified a significant synergy in the manuscript with the concept of One Conservation (https://doi.org/10.1590/1984-3143-AR2021-0024). This reviewer invites the authors to become acquainted with this concept, which could not only be explored but expanded upon in this remarkable manuscript. The concept differentiates from the already cited OnePlan, as it focuses on connecting conservation actors and seeks to "bridge the gap between wild and captive population management," understanding that it centers on parties already involved in the conservation of endangered species (wildlife conservationists and zoo/aquarium communities).

The One Conservation concept seeks to demonstrate the importance of changing paradigms among other parties not traditionally engaged in conservation programs but essential for ensuring the conservation of species and ecosystems. This interconnection, considered "an interconnection between ex situ and in situ conservation plans, anthropic actions on the environment (sustainability), and research in different areas encompassing conservation," is discussed more extensively in the article's integrated vision section. This section points out the current gap between agribusiness and the conservation community. Clear examples of where this concept is applied at the manuscript:

- Lines 472-473: "...and employing knowledge gained from advancements in domestic taxa..." – This aligns precisely with the One Conservation concept.

- Lines 588-593: An excellent example of the One Conservation concept.

Therefore, the only contribution this reviewer can offer to the authors' fantastic work is to suggest exploring the One Conservation concept further. It is important to clarify that this is not a condition for this reviewer's acceptance of the article but merely a suggestion for enriching it.

Finally, this reviewer appreciates the authors for their significant work, which will be frequently utilized and referenced by this reviewer and their group in ongoing projects involving biobanking, genetic rescue of deceased donors through tissue revitalization via xenotransplantation, and cloning wild species using domestic oocyte donors. This work will significantly contribute to this reviewer and their team.

3. Specific Comments

Line 96-97 – Line 96-97: The authors omitted Brazil, one of the countries with the highest production of bovine and equine clones today. Over 700 bovine clones have been produced in Brazil: https://globorural.globo.com/pecuaria/boi/noticia/2025/02/mais-de-700-clones-bovinos-ja-foram-produzidos-no-brasil.ghtml

182-183 – Line 182-183: The tables are not presented in the manuscript. Are these the worksheets in the Supplementary section? If so, lines 182 and 183 are confusing when checking Supplementary Sheet S2. (This difficulty led to the suggestion of a "minor review.")

Lines 438-454: Brilliant!! Congratulations on developing this paragraph!

Line 559-568: Excellent paragraph!

Author Response

Comment 1: This reviewer identified a significant synergy in the manuscript with the concept of One Conservation (https://doi.org/10.1590/1984-3143-AR2021-0024). This reviewer invites the authors to become acquainted with this concept, which could not only be explored but expanded upon in this remarkable manuscript. The concept differentiates from the already cited OnePlan, as it focuses on connecting conservation actors and seeks to "bridge the gap between wild and captive population management," understanding that it centers on parties already involved in the conservation of endangered species (wildlife conservationists and zoo/aquarium communities).

The One Conservation concept seeks to demonstrate the importance of changing paradigms among other parties not traditionally engaged in conservation programs but essential for ensuring the conservation of species and ecosystems. This interconnection, considered "an interconnection between ex situ and in situ conservation plans, anthropic actions on the environment (sustainability), and research in different areas encompassing conservation," is discussed more extensively in the article's integrated vision section. This section points out the current gap between agribusiness and the conservation community. Clear examples of where this concept is applied at the manuscript:

- Lines 472-473: "...and employing knowledge gained from advancements in domestic taxa..." – This aligns precisely with the One Conservation concept.

- Lines 588-593: An excellent example of the One Conservation concept.

Therefore, the only contribution this reviewer can offer to the authors' fantastic work is to suggest exploring the One Conservation concept further. It is important to clarify that this is not a condition for this reviewer's acceptance of the article but merely a suggestion for enriching it.

Finally, this reviewer appreciates the authors for their significant work, which will be frequently utilized and referenced by this reviewer and their group in ongoing projects involving biobanking, genetic rescue of deceased donors through tissue revitalization via xenotransplantation, and cloning wild species using domestic oocyte donors. This work will significantly contribute to this reviewer and their team.

Response 1: We thank the reviewer for bringing the One Conservation Concept to our attention. We agree with the reviewer’s emphasis on its relevance to our manuscripts major points. We have added it as a citation in the appropriate places.

Comment 2: Line 96-97 – Line 96-97: The authors omitted Brazil, one of the countries with the highest production of bovine and equine clones today. Over 700 bovine clones have been produced in Brazil: https://globorural.globo.com/pecuaria/boi/noticia/2025/02/mais-de-700-clones-bovinos-ja-foram-produzidos-no-brasil.ghtml

Response 2: We thank the reviewer for bringing attention to this glaring omission. We were familiar with the cloning work in Brazil and yet somehow managed to overlook it when listing countries that have commercial and major research cloning facilities. We have inserted Brazil into the list of nations.

Comment 3: 182-183 – Line 182-183: The tables are not presented in the manuscript. Are these the worksheets in the Supplementary section? If so, lines 182 and 183 are confusing when checking Supplementary Sheet S2. (This difficulty led to the suggestion of a "minor review.")

Response 3: We had hoped the tables would be placed into the main text. They had to be uploaded separately. We have consulted with the editor and we have inserted the table into the main text.

Reviewer 2 Report

Comments and Suggestions for Authors
  1. Brief Summary

This article aims to review the histories of commercial cloning and conservation. The authors examine the history of cloning in an attempt to understand how the divergence between views of cloning for conservation and cloning for other industries arose. They also examine the barriers that have so far prevented cloning from becoming a more widely applied conservation tool. After conducting a comprehensive literature review, they were able to create a more comprehensive list of species and subspecies that have been successfully cloned, defining success as achieving a live birth or hatching of a cloned individual. The authors conclude by demonstrating that at least one, and often multiple, healthy reproductive clones were produced for the majority of the 56 total taxa that reported live births/hatching of cloned individuals. They also report that a total of 14 taxa have been successfully cloned for conservation research and that these efforts have produced a total of 72 live births for conservation research. Of these 14 species, 38 clones (53%) from 10 taxa (71%) survived the weaning stages with fertility reported for 6 of these species. Finally, they consider low investment to be the main limitation for the expansion of cloning for conservation.

  1. General Concept Comments

This reviewer commends the authors for the topic of the article. The conservation of endangered species requires tools that allow for increased genetic variability. Furthermore, ex situ conservation will only be effective with a genetically viable population. In this sense, the use of cloning for conservation will allow genetic oxygenation between captive and free-living populations.

  1. Specific Comments

In my opinion, this article can be accepted for publication in its current form, without the need for revisions.

Author Response

We thank the reviewer for their decision to accept the manuscript as is. However, to address the other reviewer's comments we have made minor edits to insert references and add the Tables to the main text.

Reviewer 3 Report

Comments and Suggestions for Authors

The authors made an extensive and very good review in the history of the animal cloning by nuclear transfer of somatic cells. However some points need to be adressed:

  • simple summary and abstract has more words than established by the journal rules (200 words).
  • Brazilian researchers has intense relationship with cloning in farm animals, and still has active cloning companies focused on cattle and horses (i.e. InVitro Brazil, Vitrogen, Geneal, etc). Also a Governamental Research Institute (Embrapa Genetic Resources & Biotechnology) has a good history in cloning Brazilian native cattle breeds and some programs to preserve some native animals (i.e. Chrysocyon brachyurus, Mazama gouazoubira and Panthera onca).
  • Will be tables presented as supplemental files or in the manuscript? If supplemental, need to change the citation in the text.
  • Overall the writing in the text looks like more a Scientific Textbook than a Scientific Review. Several paragraphs has much of authors impression than literature, this is evidenced by several large paragraphs with any or few references, i.e.
      • lines 151 to 187 - any references
      • lines 189 to 205 - one reference
      • lines 264 to 278 - one reference
      • lines 387 to 474 - three references
      • lines 496 to 541 - four references
      • lines 559 to 621 - six references

Author Response

Comment 1: simple summary and abstract has more words than established by the journal rules (200 words).

Response 1: The simple summary is 207 words, which we believe is acceptable. Given the breadth of the paper we cannot reduce the abstract, which is currently 487 words. The editor has not flagged this as an issue.

Comment 2: Brazilian researchers has intense relationship with cloning in farm animals, and still has active cloning companies focused on cattle and horses (i.e. InVitro Brazil, Vitrogen, Geneal, etc). Also a Governamental Research Institute (Embrapa Genetic Resources & Biotechnology) has a good history in cloning Brazilian native cattle breeds and some programs to preserve some native animals (i.e. Chrysocyon brachyurus, Mazama gouazoubira and Panthera onca).

Response 2: We thank the reviewer for bringing attention to this glaring omission. We were familiar with the cloning work in Brazil and yet somehow managed to overlook it when listing countries that have commercial and major research cloning facilities. We have inserted Brazil into the list of nations.

Comment 3: Will be tables presented as supplemental files or in the manuscript? If supplemental, need to change the citation in the text.

Response 3: We had hoped the tables would be placed into the main text. They had to be uploaded separately. We have consulted with the editor and we have inserted the table into the main text.

Comment 4: Overall the writing in the text looks like more a Scientific Textbook than a Scientific Review. Several paragraphs has much of authors impression than literature, this is evidenced by several large paragraphs with any or few references, i.e.

      • lines 151 to 187 - any references
      • lines 189 to 205 - one reference
      • lines 264 to 278 - one reference
      • lines 387 to 474 - three references
      • lines 496 to 541 - four references
      • lines 559 to 621 - six references

Comment 4: For lines 151-187, the references to all the information in this paragraph are in Table 1. We should have clarified that the paragraph was outlining in text the information that is available in Table 1. It has been edited to clarify that all the information stated is supported by the references listed in Table 1. We have inserted relevant references from Table 1 into the paragraph.

For the other sections, there are few references that are relevant to the points made in these paragraphs. Most of the points that refer to the overall history of cloning, even those that seem speculative, have been published by the many previous reviews of the subject (cited in the early section of the paper). Since they are statements that have been made many times before we felt they fell into common knowledge and did not require repeated references to previous reviews. Other points in these sections are our own interpretations and therefore do not have citations. The other reviewers did not find the manuscript inappropriate in this regard, and therefore we have chosen not to insert additional references here.

Round 2

Reviewer 3 Report

Comments and Suggestions for Authors

The presented revised version of the manuscript had some minor changes, and the major revision made before was not considered by the authors. The language of the manuscript still majorly as opinions, that is suitable for a textbook, not for a scientific review.

Comment 1: was completed unconsidered 

Comment 2: just word Brazil was added in the text

Comment 3: was majorly unconsidered.

Author Response

Comment 1: simple summary and abstract has more words than established by the journal rules (200 words).

Response 1: We have now shortened the simple summary to 195 words and the abstract has been reduced from 487 words to 261 words, which we recognize is still above the 200 word limit but now allows both the simple summary and abstract to fit on the first page of the manuscript.

Comment 2: just word Brazil was added in the text (text from round 1 review - Brazilian researchers has intense relationship with cloning in farm animals, and still has active cloning companies focused on cattle and horses (i.e. InVitro Brazil, Vitrogen, Geneal, etc). Also a Governamental Research Institute (Embrapa Genetic Resources & Biotechnology) has a good history in cloning Brazilian native cattle breeds and some programs to preserve some native animals (i.e. Chrysocyon brachyurus, Mazama gouazoubira and Panthera onca).

Response 2: The reviewer will note that our paper did not provide extensive detail for the extent of cloning activities in any nation - no companies, institutes, or universities were explicitly named for any other country. The nations in which cloning activities are performed were simply listed to show the global nature of the industry. 

Comment 3: was majorly unconsidered.

Response 3: We believe the reviewer is referring to the previous comment 4, regarding the “textbook” nature of the paper and paragraphs with few references. We have gone through several statements in the discussion and modified them to either remove the speculative comments or make it clear that the comments are the author’s and not based in previously reported papers. We have added several relevant and supporting references for our hypotheses and seemingly more conjectural statements.